# Role of hematological parameters in the early detection of clinical cases for septicemia among neonates: A hospital-based study from Chennai, India

**Jeivanth S.B.**[1], **Shreemathee Baskar**[1], **Mohammad Fareed**[2]*, **K. Santhosh Kumar**[3], **Osama Elshahat Mostafa**[4], **Amen Bawazir**[2], **Khalid I. AlQumaizi**[5]

**1** Saveetha Medical College and Hospital, Saveetha Institute of Medical and Technical Sciences (SIMATS) Chennai, India, **2** Community Medicine Unit, Department of Basic Medical Sciences, College of Medicine, AlMaarefa University, Diriyah, Saudi Arabia, **3** Department of Neonatology, Saveetha Medical College and Hospital, Saveetha Institute of Medical and Technical Sciences (SIMATS) Chennai, India, **4** Department of Nursing, College of Applied Sciences, AlMaarefa University, Diriyah, Saudi Arabia, **5** Department of Family Medicine, College of Medicine, AlMaarefa University, Diriyah, Saudi Arabia

* fareed.research@gmail.com

## Abstract

### Background

Neonatal sepsis, a leading cause of newborn mortality, arises from systemic infections due to an immature immune system. Its subtle early symptoms complicate timely diagnosis. Hematological parameters act as an indicator for early detection, crucial for prompt treatment, improving prognosis, and are not a challenging or cumbersome process.

### Aim

The primary objective was to evaluate the significance of hematological parameters including red blood cell (RBC), WBC, and platelet counts in the context of neonatal sepsis.

### Methods

This hospital-based cohort study examined 73 neonates admitted to the neonatal intensive care unit (NICU) of Saveetha Medical College and Hospital, Chennai, India during the period of January 2023 to March 2024. All the new born patients were presented with blood culture-confirmed septicemia.

### Results

The investigation identified *Klebsiella pneumoniae* as the most prevalent etiological agent (26.02%), followed by Coagulase-Negative Staphylococci (CONS) and *Acinetobacter baumannii* (both 8.2%). Alterations in total leukocyte count and hematocrit were observed in 57% and 68.1% of cases, respectively, providing a prompt indication of infection status. Subsequent analyses revealed prominent leukocytosis, hematocrit irregularities, and

**Data availability statement:** Public sharing of the data is restricted in accordance with the policies of the Neonatal Intensive Care Unit (NICU) at Saveetha Medical College and Hospital. Permission for data usage has been granted by the hospital administration strictly for research purposes, adhering to institutional regulations and ethical standards. Below is the information of non-author for data access and ethics committee so that other researchers can request access to our data: 1. Dr. Deepak Nallaswamy Pro-Chancellor and Director, Saveetha Medical College and Hospital, Chennai, India E-mail: dir.acad@ saveetha.com 2. Dr. Nisha B. Associate Dean for Research, Head of Scientific Review Board (Ethics Committee), Saveetha Medical College and Hospital Chennai, India E-mail: nishachandru21@gmail.com.

**Funding:** The author(s) received no specific funding for this work.

**Competing interests:** The authors have declared that no competing interests exist.

**Abbreviations:** Hct, Hematocrit; RBC, Red Blood cell; MCV, Mean corpuscular volume; MCH, Mean corpuscular hemoglobin; MCHC, Mean corpuscular hemoglobin concentration; WBC, White Blood cell; TLC, Total Leucocyte Count; ANC, Absolute Neutrophil Count ; ALC, Absolute Lymphocyte Count; NBW, Normal Birth Weight; LBW, Low Birth Weight; VLBW, Very Low Birth Weight

thrombocytopenia, frequently manifesting in septic cases and demonstrating potential as early markers for neonatal sepsis.

## Conclusion

The study highlights the diagnostic value of hematological alterations, such as leukocytosis and hematocrit distortion, in the prompt identification of septicemia among neonates. Based on the findings, it is recommended that routine hematological screening to be integrated as a standard component of neonatal sepsis diagnosis for rapid investigation of neonatal sepsis.

## 1. Introduction

Neonatal sepsis is a systemic infectious condition that occurs in neonates from 0–28 days of life [1] and is one of the leading causes of morbidity and fatality [2]. Early-onset neonatal sepsis (EOS) is characterized by age at the onset, with bacteremia or bacterial meningitis occurring at < 72 h in newborns being hospitalized in the Neonatal Intensive Care Unit (NICU), late-onset sepsis (LOS) is a form of sepsis that develops in NICU neonates after 72 hours [3]. It has also been variously reported as happening up to a maximum age of 90 or 120 days and may be caused by either horizontally or vertically acquired illnesses. Early-onset newborn infections of viral or fungal origin may also arise at seven days of life and must be differentiated from bacterial sepsis, for instance, viral infections, including those caused by emerging and re-emerging neurotropic viruses such as Zika virus, enterovirus, and parechovirus, can present with central nervous system involvement and symptoms overlapping bacterial sepsis. These infections require precise diagnostic approaches to ensure timely and appropriate management [4]. It has been estimated that 2202 per 100,000 live births were proven to have sepsis, with a mortality rate of 11–19% around the world [5]. The prevalence of neonatal sepsis in India is 38 per 1000 intramural live births, and septicemia is the most frequent, with a frequency of 24 per 1000 live births, accounting for a mortality rate of 23% [6]. Timely detection of neonatal sepsis is critical for commencing adequate therapy and avoiding severe outcomes [7]. However, present diagnostic procedures have limits, prompting the discovery of alternative ways to increase accuracy and efficiency. Blood culture remains the gold standard for diagnosing sepsis, providing definitive microbiological evidence to guide targeted treatment. However, it is a time-consuming process, requiring 24–48 hours or more for results, and necessitates a well-equipped laboratory with appropriate technical expertise. The delay in obtaining results can lead to significant challenges in clinical management, particularly in critical cases where timely intervention is essential [8,9]. The parameters predicting neonatal sepsis are neutropenia, thrombocytopenia [2], and anemia [10]. In infants, one of the primary causes of thrombocytopenia is sepsis, and thrombocytopenia could swiftly become quite severe within 24–48 hours following the onset of the infection [11]. It has been postulated that damage to the endothelium occurs in neonatal sepsis, which stimulates the reticuloendothelial clearance of platelets.

The diagnosis of neonatal sepsis presents particular difficulties due to vague clinical presentations and newborns' inability to express symptoms [12]. Moreover, standard diagnostic procedures such as blood culture have limitations in terms of sensitivity, specificity, and turnaround time. Delayed or delayed diagnosis may lead to septic shock, multiorgan failure, and death [13]. Therefore, there is a significant need for accurate and quick diagnostic methods to identify newborns at risk of sepsis and commence prompt therapies.

Conventional diagnostic techniques for newborn sepsis, including blood culture and laboratory indicators, which include C-reactive protein (CRP) and procalcitonin (PCT), have significant limitations. Blood culture, considered the gold standard, is hindered by poor sensitivity, protracted turnaround time, and the demand for substantial blood volume, which may not be practicable in critically ill neonates [14]. CRP and PCT lack acceptable sensitivity and specificity, especially in the early stages of infection [15]. As a result, there is an urgent need for improved diagnostic techniques to overcome these obstacles and increase neonatal sepsis identification accuracy and timeliness. Early identification and rapid treatment are critical in avoiding the development of severe sepsis and septic shock. Hematological markers, including WBC, neutrophil-to-lymphocyte ratio (NLR), platelet count, and coagulation parameters, give useful insights into the inflammatory and coagulation cascades associated with sepsis [16].

Alterations in WBC count, such as leukocytosis or leukopenia, are typically reported in sepsis. Neutropenia, or leukopenia, is an especially dangerous indicator of the severity of sepsis [17]. While a high WBC count may signal infection, it lacks specificity and may be altered by different circumstances, limiting its diagnostic utility [18]. Evaluation of the differential counts, especially neutrophil and lymphocyte percentages, might offer insights into the inflammatory response associated with sepsis. A higher neutrophil-to-lymphocyte ratio (NLR) has been associated with sepsis severity and poor outcomes, indicating its potential as a prognostic biomarker. Anaemia is a common symptom in septic patients and can come from a variety of causes, including inflammation, hemolysis, and blood loss. Monitoring hemoglobin levels may assist in determining the severity of sickness and directing transfusion options in septic patients. Thrombocytopenia is typically detected in sepsis and is symptomatic of coagulopathy and microvascular dysfunction. Monitoring platelet count may assist in identifying individuals at risk of sepsis-associated sequelae, such as disseminated intravascular coagulation (DIC) [19]. Among the organisms causing neonatal sepsis, Klebsiella pneumoniae is the most typically isolated bacteria, followed by Staphylococcus aureus and Pseudomonas aeruginosa in many studies [2]. This research seeks to investigate the diagnostic significance of hematological markers in sepsis and their prospective repercussions for therapeutic care.

This study aims to analyze the hematological and culture findings in neonates with septicemia, to study the alterations in the blood counts, namely, WBC, RBC, and platelet count, in proven cases of neonatal sepsis, to study the alterations in the differential count, which includes neutrophils, lymphocytes, monocytes, eosinophils, and basophils, and to investigate the organisms that cause sepsis in neonates.

## Materials and Methods

This is a small cohort study that was conducted on the hematological data obtained from neonatal patients with sepsis confirmed by culture. This study has received ethical approval from Scientific Review Board of Saveetha Medical College and Hospital, Chennai, India. The ethical approval number for this study is: 413/11/2024/UG/SRB/SMCH. Informed verbal consent was obtained from the parents or legal guardians of the neonates for the use of patient data solely for research purposes. The consent process was carried out in alignment with ethical standards, ensuring that the parents or guardians were thoroughly informed about the study's objectives, procedures, and scientific benefits. The verbal consent was duly documented in the patient records and corroborated by the presence of a healthcare professional as a witness. Additionally, the study protocol, including the consent procedure, was reviewed and approved by the Scientific Review Board of Saveetha Medical College and Hospital, Chennai, ensuring compliance with institutional and international ethical guidelines for research involving human participants.

The details of neonates with positive cultures were obtained from the records of the NICU and laboratory of Saveetha Medical College and Hospital. The hemogram details were obtained from the hospital hematology lab records. Alterations in the hemogram of neonatal sepsis were tabulated and analyzed. The organisms causing this infection were studied.

The sampling used was complete enumeration sampling. This study group consisted of 73 neonates aged 0-28 days with culture-proven sepsis, during the period of January 2023 to March 2024. The inclusion criteria for the study were that neonates (28 days old), with culture-proven septicemia and neonates without septicemia were excluded from the study. RBC parameters such as RBC count, hematocrit, MCV, MCH, MCHC, and hemoglobin levels, and WBC parameters such as total and differential leucocyte count and platelet count were taken into account. The demographic details were collected from the NICU at Saveetha Medical College and Hospital. Data on cases of culture-positive sepsis were obtained from the microbiology laboratory. The corresponding hematological findings in the culture-positive cases were taken from the hematology laboratory records. The patients were classified by their age, sex, and causative organisms, and further by whether the blood culture was positive at early onset or at late onset.

Finally, they were classified into three groups by their weight: Normal ($\geq 2,500\,g$); low birth weight (LBW) ($1,500–2,500\,g$); and very low birth weight (VLBW) ($\leq 1,500\,g$). The results were tabulated and analyzed using statistical software, SPSS version 29.0.2.0 for Windows. The reference values for hematological profiles among neonates were taken from the Nelson Textbook of Pediatrics, 21st ed [14]., and OP Ghai Essential Pediatrics [20], 9th ed., PG Textbook of Pediatrics [21] which are shown in Table 1.

## Results

### 1. Demographic and physical characteristics of neonatal sepsis cases

Table 2 provides a detailed overview of the demographic and physiological characteristics of neonates diagnosed with sepsis. Among the 73 cases, the distribution by sex reveals that males were more frequently affected, accounting for 63.0% of the cases, compared to females at 37.0%. Regarding gestational age, a larger proportion of cases were preterm infants, comprising 53.4%, while term infants represented 46.6%. This indicates a higher susceptibility to sepsis among preterm neonates, likely due to their underdeveloped immune systems. The cases were further categorized based on birth weight, with 46.6% of neonates classified as having normal birth weight (NBW), 23.3% as low birth weight (LBW), and 30.1% as very low birth weight (VLBW). The high percentage of cases in the VLBW group further underscores the increased vulnerability of lower birth weight infants to infections, which is critical for guiding clinical approaches in managing and preventing neonatal sepsis in these high-risk groups.

### 2. Distribution of cases based on their age

Table 3 demonstrates a breakdown of neonatal sepsis cases according to the day of onset, highlighting the most common age at which sepsis was observed in neonates. The data reveals that the majority of cases occurred on the day of birth (23.3%), highlighting that sepsis onset is most frequently identified in neonates shortly after birth. Following this, smaller percentages were observed in the days immediately after birth, with a gradual decline in frequency over time. For instance, 11.0% of cases occurred on Day 5 (D5), and 9.6% on Day 3 (D3), indicating higher occurrences within the first week. Beyond the first week, the frequency of cases progressively declined, with only 1.4% recorded on days such as D6, D12, and D20, reflecting minimal occurrences. This distribution underscores the heightened susceptibility to sepsis immediately after birth, with a decreasing likelihood of onset as the neonate's age increases.

**Table 1. Reference values for for Neonates.**

| Parameter | Reference Range |
|---|---|
| Red Blood Cell Parameters | |
| Hemoglobin (Hb, g/dl) | 15-24 |
| Hematocrit (Hct, %) | 44–70 |
| RBC ($*10^{12}$/L) | 5.3-5.6 |
| Mean Corpuscular Volume (MCV, fl) | 99-115 |
| Mean Corpuscular Hemoglobin (MCH, pg) | 33-39 |
| Mean Corpuscular Hemoglobin Concentration (MCHC, g/dl) | 32-36 |
| Leukocyte Parameters | |
| Total Leukocyte Count (TLC, cells/mm³) | 9100-34,000 |
| Neutrophils (seg, %) | 54-62 |
| Lymphocytes (%) | 25-33 |
| Monocytes (%) | 3-7 |
| Eosinophils (%) | 1-3 |
| Basophils (%) | 0-0.75 |
| **Platelet Count ($10^6$ cells/mm³)** | |
| At birth | 84-478 |
| 1 week - adult | 150-400 |
| Gestational Age (weeks) | |
| Term Baby | 37-42 |
| Preterm Baby | <37 |
| Post-term Baby | >42 |
| Birth Weight (grams) | |
| Normal Birth Weight (NBW) | 2500-4000 |
| Low Birth Weight (LBW) | 1500-2499 |
| Very Low Birth Weight (VLBW) | 1000-1499 |

**Table 2. Characteristics of Neonatal Sepsis Cases Based on Sex, Gestational Age, and Birth Weight.**

| Characteristic | Category | n (n %) |
|---|---|---|
| Sex | Female | 27(37.0) |
| | Male | 46 (63.0) |
| Gestational Age | Preterm | 39 (53.4) |
| | Term | 34 (46.6) |
| Birth Weight | Very Low Birth Weight | 22 (30.1) |
| | Low Birth Weight | 17 (23.3) |
| | Normal Birth Weight | 34 (46.6) |

These findings emphasize the importance of close monitoring, particularly on the day of birth and during the early neonatal days.

## 3. RBC profile of neonatal septicemia cases

Table 4 presents the hematological profile focusing on RBC parameters in cases of neonatal sepsis across various days of onset. Hemoglobin (Hb) levels, hematocrit (Hct), RBC count, mean corpuscular volume (MCV), mean corpuscular hemoglobin (MCH), and mean corpuscular hemoglobin concentration (MCHC) were measured. On the day of birth (D0),

**Table 3. Distribution of cases based on age in days.**

| Age (in days) | n (n %) |
|---|---|
| D0 | 17 (23.30) |
| D1 | 2 (2.70) |
| D2 | 6 (8.20) |
| D3 | 7 (9.60) |
| D4 | 6 (8.20) |
| D5 | 8 (11.0) |
| D6 | 1 (1.40) |
| D7 | 5 (6.80) |
| D8 | 3 (4.10) |
| D9 | 3 (4.10) |
| D10 | 2 (2.70) |
| D11 | 2 (2.70) |
| D12 | 1 (1.40) |
| D13 | 1 (1.40) |
| D14 | 1 (1.40) |
| D16 | 2 (2.70) |
| D18 | 2 (2.70) |
| D19 | 1 (1.40) |
| D20 | 1 (1.40) |
| D25 | 1 (1.40) |
| D28 | 1 (1.40) |

neonates with sepsis had an average Hb level of 14.47 g/dL, with a hematocrit of 43.05% and an RBC count of 4.48 * $10^{12}$/L. MCV and MCH values generally fluctuated across the days, showing variation in cell size and hemoglobin content among neonates. The highest Hb was noted on D1 with 17.55 g/dL, while values for both Hb and Hct tended to decrease as the days progressed, particularly evident by D25 and D28. Standard deviations reflect a considerable variability in these parameters among cases, likely due to individual differences in response to infection and other clinical factors affecting RBCs during the early neonatal period.

The present study found that neonates with septicemia on day 0 accounted for 23.3% of cases, representing the most common occurrence. Male neonates were more frequently affected (63.0%) compared to females (37.0%). Regarding birth weight, 46.6% of neonates had normal birth weight, 23.3% had low birth weight, and 30.1% were classified as very low birth weight. In terms of gestational age, 53.4% of cases were preterm infants, while 46.6% were term infants. This distribution underscores the higher occurrence of neonatal sepsis in preterm infants, reflecting their increased susceptibility to infections compared to term neonates.

## 4. WBC profile of neonatal septicemia cases

Table 5 summarizes the WBC profiles of neonates with septicemia across different ages of onset. The data shows that on the day of birth (D0), the mean Total Leucocyte Count (TLC) was 12,636.47 cells/mm³, with neutrophils making up about 49.8% and lymphocytes about 37.8%, indicating a mixed cellular response. The ANC averaged 6,671.24, while the ALC was slightly lower at 6,219.41, reflecting a higher proportion of neutrophils in response to infection. Platelet counts on D0 averaged $2.59 \times 10^{6}$ cells/mm³, showing moderate levels,

**Table 4. RBC profile of neonatal septicemia cases.**

| Age of Onset in Days(D) | | Hb(g/dl) | Hct(%) | RBC($*10^{12}$/L) | MCV(fl) | MCH(pg) | MCHC(g/dl) |
|---|---|---|---|---|---|---|---|
| D0 (n = 17) | Mean | 14.471 | 43.053 | 4.4818 | 97.941 | 32.935 | 33.606 |
| | Std. Dev | 3.6213 | 10.5326 | 1.12166 | 14.9321 | 5.1046 | 1.5642 |
| D1 (n = 2) | Mean | 17.55 | 50.65 | 5.04 | 101.15 | 35.05 | 34.7 |
| | Std. Dev | 0.6364 | 2.8991 | 0.72125 | 8.6974 | 3.7477 | 0.7071 |
| D2 (n = 6) | Mean | 16.7 | 47.817 | 5.3583 | 89.35 | 31.167 | 34.85 |
| | Std. Dev | 2.1753 | 5.2537 | 0.45243 | 8.2936 | 3.4139 | 0.905 |
| D3 (n = 7) | Mean | 16.414 | 48.357 | 4.6114 | 104.443 | 35.457 | 34.029 |
| | Std. Dev | 3.4523 | 10.5143 | 0.87564 | 6.008 | 1.8937 | 0.905 |
| D4 (n = 6) | Mean | 15.467 | 44.117 | 4.3783 | 101.55 | 35.517 | 35 |
| | Std. Dev | 2.7998 | 7.5529 | 0.90172 | 5.1208 | 1.5289 | 0.9757 |
| D5 (n = 8) | Mean | 14.625 | 42.113 | 4.1938 | 100.15 | 34.788 | 34.725 |
| | Std. Dev | 3.2186 | 9.3291 | 0.82784 | 6.2555 | 2.2762 | 0.6964 |
| D6 (n = 1) | Mean | 18.7 | 58.1 | 5.58 | 104.1 | 33.5 | 32.2 |
| | Std. Dev | 0.00 | 0.00 | 0.00 | 0.00 | 0.00 | 0.00 |
| D7 (n = 5) | Mean | 15.74 | 45.68 | 4.462 | 102.86 | 35.4 | 34.44 |
| | Std. Dev | 2.1698 | 5.8713 | 0.65964 | 8.1871 | 2.5739 | 0.5273 |
| D8 (n = 3) | Mean | 13.533 | 39.267 | 3.95 | 101.033 | 34.733 | 34.733 |
| | Std. Dev | 3.5726 | 11.0192 | 1.29723 | 15.7906 | 2.3587 | 3.8527 |
| D9 (n = 3) | Mean | 18.433 | 53.033 | 5.2667 | 101.1 | 35.233 | 34.933 |
| | Std. Dev | 3.6828 | 12.1517 | 1.17798 | 9.6768 | 2.4947 | 1.3614 |
| D10 (n = 2) | Mean | 14 | 41 | 4.025 | 105.15 | 35.85 | 34.15 |
| | Std. Dev | 4.2426 | 12.3037 | 1.74655 | 15.0614 | 5.0205 | 0.0707 |
| D11 (n = 2) | Mean | 14.55 | 41.45 | 4.175 | 99.3 | 34.85 | 35.1 |
| | Std. Dev | 1.9092 | 5.5861 | 0.57276 | 0.2828 | 0.2121 | 0.1414 |
| D12 (n = 1) | Mean | 9.5 | 28 | 2.82 | 99.3 | 33.7 | 33.9 |
| | Std. Dev | 0.00 | 0.00 | 0.00 | 0.00 | 0.00 | 0.00 |
| D13 (n = 1) | Mean | 11.8 | 32.9 | 3.57 | 92.2 | 33.1 | 35.9 |
| | Std. Dev | 0.00 | 0.00 | 0.00 | 0.00 | 0.00 | 0.00 |
| D14 (n = 1) | Mean | 11 | 30.8 | 3.34 | 92.2 | 32.9 | 35.7 |
| | Std. Dev | 0.00 | 0.00 | 0.00 | 0.00 | 0.00 | 0.00 |
| D16 (n = 2) | Mean | 16.65 | 47.9 | 4.7 | 102.05 | 35.25 | 34.5 |
| | Std. Dev | 7.8489 | 21.3546 | 2.12132 | 0.6364 | 0.7778 | 0.9899 |
| D18 (n = 2) | Mean | 14.55 | 41.4 | 4.425 | 92.2 | 32.5 | 35.35 |
| | Std. Dev | 4.4548 | 14.0007 | 0.67175 | 17.6777 | 5.0912 | 1.2021 |
| D19 (n = 1) | Mean | 15.3 | 42.2 | 4.69 | 90 | 32.6 | 36.3 |
| | Std. Dev | 0.00 | 0.00 | 0.00 | 0.00 | 0.00 | 0.00 |
| D20 (n = 1) | Mean | 10 | 28.2 | 2.88 | 97.9 | 34.7 | 35.5 |
| | Std. Dev | 0.00 | 0.00 | 0.00 | 0.00 | 0.00 | 0.00 |
| D25 (n = 1) | Mean | 8.8 | 24.9 | 2.56 | 97.3 | 34.4 | 35.3 |
| | Std. Dev | 0.00 | 0.00 | 0.00 | 0.00 | 0.00 | 0.00 |
| D28 (n = 1) | Mean | 12.7 | 40.8 | 3.78 | 107.9 | 33.6 | 31.1 |
| | Std. Dev | 0.00 | 0.00 | 0.00 | 0.00 | 0.00 | 0.00 |

though some days displayed marked variations. In 23.3% of cases, the total leukocyte count remained positive, while in most cases, the ANC count stayed normal. Neutrophil and lymphocyte abnormalities were noted in 85.3% and 86.3% of cases, respectively, indicating a high

**Table 5. WBC profile of neonatal septicemia cases.**

| Age of Onset in Days(D) | | TLC (cells/mm³) | Neutrophil (%) | ANC | Lymphocyte (%) | ALC | Monocyte (%) | Eosinophil (%) | Basophil (%) | Platelet Count (10⁶cells/mm³) |
|---|---|---|---|---|---|---|---|---|---|---|
| D0 (n=17) | Mean | 12636.47 | 49.812 | 6671.24 | 37.794 | 6219.41 | 5.7 | 2.776 | 0.347 | 2.5853 |
| | Std. Dev | 5876.915 | 19.0373 | 4379.711 | 18.7713 | 3177.871 | 2.781 | 4.0905 | 0.2649 | 1.23222 |
| D1 (n=2) | Mean | 10535 | 65.4 | 5185 | 24.35 | 4037.5 | 6.2 | 2.4 | 1.65 | 0.98 |
| | Std. Dev | 12141.023 | 30.1227 | 4574.981 | 24.2538 | 5511.897 | 2.2627 | 3.3941 | 0.2121 | 0.93338 |
| D2 (n=6) | Mean | 15778.33 | 70 | 11001.67 | 20.55 | 3064.33 | 5.617 | 2.233 | 1.6 | 2.0167 |
| | Std. Dev | 8851.213 | 11.2011 | 6145.299 | 8.9337 | 1714.522 | 2.9267 | 2.121 | 1.7675 | 1.61335 |
| D3 (n=7) | Mean | 9101.43 | 60.143 | 5909 | 31.143 | 2623 | 6.771 | 1.543 | 0.771 | 2.3029 |
| | Std. Dev | 6480.41 | 16.5766 | 4577.641 | 18.2484 | 1680.001 | 2.2831 | 1.0196 | 0.4751 | 0.4172 |
| D4 (n=6) | Mean | 8578.33 | 52.55 | 5298.33 | 32.717 | 2667.67 | 11.033 | 3.15 | 0.55 | 1.675 |
| | Std. Dev | 5299.428 | 21.1673 | 4755.428 | 17.2718 | 1323.47 | 8.0286 | 3.1513 | 0.3728 | 0.93695 |
| D5 (n=8) | Mean | 9427.5 | 66.125 | 6233.38 | 24.95 | 2379.5 | 5.963 | 2.25 | 0.713 | 1.7863 |
| | Std. Dev | 5165.439 | 16.5786 | 3921.098 | 13.6786 | 2186.161 | 4.1562 | 1.069 | 0.8442 | 1.4634 |
| D6 (n=1) | Mean | 9350 | 91.8 | 8580 | 3 | 281 | 4.6 | 0.3 | 0.3 | 1.17 |
| | Std. Dev | 0.00 | 0.00 | 0.00 | 0.00 | 0.00 | 0.00 | 0.00 | 0.00 | 0.00 |
| D7 (n=5) | Mean | 10250 | 38.92 | 4736.8 | 48.66 | 4139.2 | 5.68 | 5.98 | 0.76 | 2.958 |
| | Std. Dev | 6066.712 | 23.2846 | 5716.406 | 20.9452 | 1964.002 | 3.1092 | 8.2163 | 0.5595 | 1.43832 |
| D8 (n=3) | Mean | 16150 | 55.2 | 8950 | 36.267 | 5716.33 | 6.867 | 1.433 | 0.233 | 0.7767 |
| | Std. Dev | 9283.701 | 7.5505 | 5210.864 | 8.0308 | 3121.697 | 2.5106 | 1.1372 | 0.1155 | 0.71009 |
| D9 (n=3) | Mean | 20323.33 | 61.033 | 12736.67 | 24.267 | 4799.33 | 8.967 | 5.2 | 0.533 | 2.4833 |
| | Std. Dev | 5977.511 | 19.7414 | 6792.749 | 8.8737 | 1585.537 | 2.7755 | 8.4006 | 0.2082 | 0.95521 |
| D10 (n=2) | Mean | 9435 | 57.9 | 5807.5 | 29.85 | 2443.5 | 11 | 0.75 | 0.5 | 1.515 |
| | Std. Dev | 2736.503 | 25.0316 | 3949.191 | 27.2236 | 1751.503 | 3.677 | 1.0607 | 0.4243 | 0.44548 |
| D11 (n=2) | Mean | 15665 | 59.3 | 9905 | 29.4 | 4219.5 | 7.5 | 3.25 | 0.55 | 4.375 |
| | Std. Dev | 8152.941 | 15.4149 | 7247.845 | 9.4752 | 912.875 | 1.9799 | 3.6062 | 0.3536 | 0.36062 |
| D12 (n=1) | Mean | 10250 | 72.2 | 7400 | 14.6 | 1497 | 7.4 | 5.3 | 0.5 | 4.75 |
| | Std. Dev | 0.00 | 0.00 | 0.00 | 0.00 | 0.00 | 0.00 | 0.00 | 0.00 | 0.00 |
| D13 (n=1) | Mean | 14400 | 38.4 | 5530 | 45.5 | 6552 | 9.6 | 5.3 | 1.2 | 5.12 |
| | Std. Dev | 0.00 | 0.00 | 0.00 | 0.00 | 0.00 | 0.00 | 0.00 | 0.00 | 0.00 |
| D14 (n=1) | Mean | 22570 | 65.2 | 14720 | 27.2 | 6139 | 6.5 | 0.8 | 0.3 | 4.87 |
| | Std. Dev | 0.00 | 0.00 | 0.00 | 0.00 | 0.00 | 0.00 | 0.00 | 0.00 | 0.00 |
| D16 (n=2) | Mean | 19515 | 67.9 | 14410.5 | 19.85 | 3292 | 7.45 | 3.4 | 1.4 | 2.335 |
| | Std. Dev | 12183.45 | 18.809 | 11963.54 | 9.5459 | 555.786 | 3.7477 | 3.8184 | 1.6971 | 0.78489 |
| D18 (n=2) | Mean | 14140 | 59.25 | 8241.5 | 27.2 | 4285 | 9.85 | 3 | 0.7 | 4.175 |
| | Std. Dev | 10521.749 | 2.6163 | 5866.865 | 8.3439 | 4041.822 | 4.4548 | 0.8485 | 0.4243 | 0.89803 |
| D19 (n=1) | Mean | 12230 | 30.4 | 3720 | 57 | 6971 | 8.1 | 3.8 | 0.7 | 6.13 |
| | Std. Dev | 0.00 | 0.00 | 0.00 | 0.00 | 0.00 | 0.00 | 0.00 | 0.00 | 0.00 |
| D20 (n=1) | Mean | 10340 | 48.9 | 5060 | 35.9 | 3712 | 13.7 | 1.3 | 0.2 | 4.38 |
| | Std. Dev | 0.00 | 0.00 | 0.00 | 0.00 | 0.00 | 0.00 | 0.00 | 0.00 | 0.00 |
| D25 (n=1) | Mean | 9170 | 36.6 | 3350 | 58 | 5319 | 5.3 | 0.1 | 0 | 3 |
| | Std. Dev | 0.00 | 0.00 | 0.00 | 0.00 | 0.00 | 0.00 | 0.00 | 0.00 | 0.00 |
| D28 (n=1) | Mean | 7490 | 17 | 1270 | 75.7 | 5670 | 4.8 | 1.7 | 0.8 | 0.93 |
| | Std. Dev | 0.00 | 0.00 | 0.00 | 0.00 | 0.00 | 0.00 | 0.00 | 0.00 | 0.00 |

incidence of immune cell irregularities among the affected neonates. Thrombocytopenia was seen in 23.3% of instances, highlighting a common but variable response to infection affecting platelet counts. Hematocrit levels remained positive in 50% of cases, with mean corpuscular

volume (MCV) changes observed in 45.2%, indicating shifts in red cell indices potentially linked to disease severity and progression. As the days progressed, there were noticeable fluctuations in these values. For example, TLC peaked on D14 at 22,570 cells/mm³, with a very high ANC of 14,720 and a relatively lower ALC, which suggests a strong neutrophil response typical in severe infection. In contrast, on D28, the TLC was lowest at 7,490 cells/mm³, with a high lymphocyte percentage (75.7%), indicating a shift from a neutrophilic to a lymphocytic response over time in some cases.

## 5. Distribution of micro-organisms based on age and birth weight

Table 6 illustrates the distribution of microorganisms causing neonatal sepsis based on the age of onset and birth weight, respectively. Regarding distribution based on age, *Klebsiella pneumoniae* is the most prevalent organism, appearing in both early-onset (9 cases) and late-onset (10 cases) sepsis, followed by *CONS* (Coagulase-negative Staphylococci) and *Acinetobacter baumannii*, which are also found in both categories. Late-onset cases tend to show greater microbial diversity, including *Pseudomonas aeruginosa* and *Serratia marcescens*.

Fig 1 displays the distribution of microorganisms isolated in cases of neonatal sepsis. Each colored segment represents a different microorganism, with the legend identifying them by name. *Klebsiella pneumoniae* appears to occupy one of the largest segments, suggesting it is one of the most prevalent pathogens in this sample. Other notable microorganisms include

**Table 6. Distribution of micro-organism based on the age of onset.**

| Micro-organisms | Early onset (n) | Late onset (n) |
|---|---|---|
| **Gram-Positive** | | |
| Candida species | 1 | 1 |
| Coagulase negative Staphlyococcus | 3 | 3 |
| Enterococcus faecalis | 1 | 1 |
| Enterococcus faecium | 0 | 2 |
| Micrococci | 3 | 1 |
| Staphylococcus aureus | 0 | 1 |
| Staphylococcus epidermidis | 1 | 2 |
| Staphylococcus haemolyticus | 3 | 0 |
| Staphylococcus hominis | 1 | 0 |
| Staphylococcus saprophyticus | 1 | 1 |
| Staphylococcus warneri | 0 | 1 |
| Streptococcus | 0 | 1 |
| **Gram-Negative** | | |
| Achromobacter xylosoxidans | 1 | 0 |
| Acinetobacter baumannii | 4 | 2 |
| Burkholderia cepacia | 1 | 0 |
| Elizabethkingia meningoseptica | 1 | 2 |
| Enterobacter aerogenes | 1 | 0 |
| Enterobacter cloacae | 1 | 2 |
| Escherichia coli | 0 | 2 |
| Klebsiella pneumoniae | 9 | 10 |
| Pseudomonas aeruginosa | 0 | 5 |
| Serratia marcescens | 0 | 4 |
| **TOTAL** | 32 | 41 |

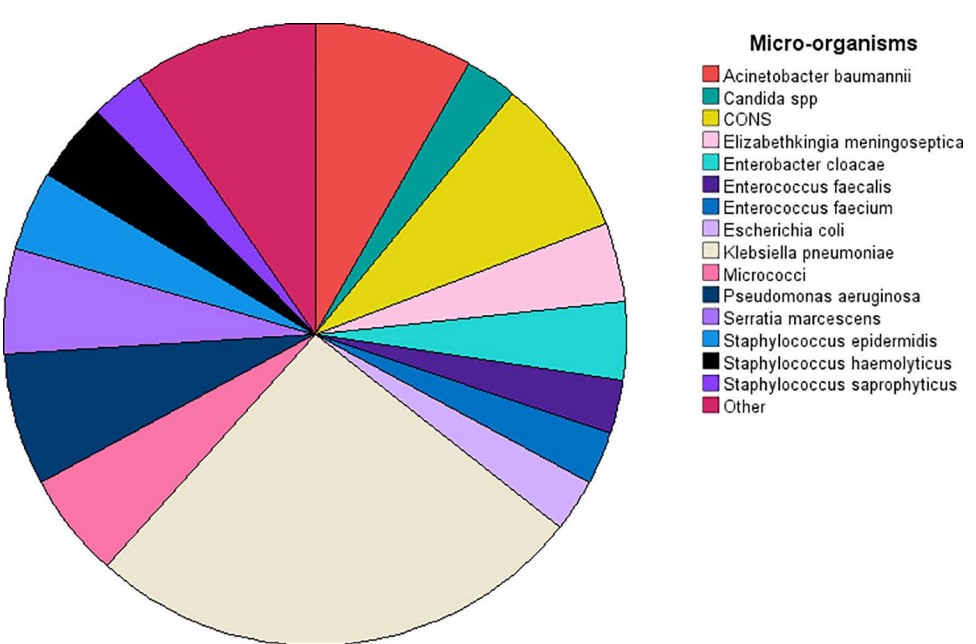

**Fig 1. Frequency of various micro-organisms isolated from the cases of septicemia.**

*Acinetobacter baumannii*, *CONS* (Coagulase-negative Staphylococci), *Pseudomonas aeruginosa*, and *Serratia marcescens*, each contributing smaller segments to the overall distribution. The "Other" category, represented by the largest beige segment, likely includes a variety of less frequent organisms, indicating a broader range of pathogens beyond the primary ones displayed.

Table 7 exhibits the distribution based on birth weight; *Klebsiella pneumoniae* remains the predominant pathogen across all birth weight groups, with 6 cases in VLBW, 4 in LBW, and 9 in NBW. Other notable organisms in the VLBW group include *Elizabethkingia meningoseptica* and *Pseudomonas aeruginosa*, suggesting a higher risk of certain infections in lower birth weight neonates.

## 6. ANOVA results for hematological profile

Table 8 displays the ANOVA results for the RBC profile in neonatal septicemia cases based on the age of onset of sepsis. There was no statistically significant difference in the RBC hematological parameters between groups, as demonstrated by one-way ANOVA analysis for hemoglobin (g/dl), ($F_{(20,52)} = 1.06$, p = .408), RBC (*10^12/L), ($F_{(20,52)} = 1.12$, p = .352), hematocrit ($F_{(20,52)} = 1.11$, p = .369), mean corpuscular volume (fl), ($F_{(20,52)} = 0.60$, p = .894), mean corpuscular hemoglobin (pg), ($F_{(20,52)} = 0.55$, p = .929), mean corpuscular hemoglobin concentration (g/dl), ($F_{(20,52)} = 1.26$, p = .242), and red cell distribution ($F_{(20,52)} = 1.65$, p = .075).

Table 9 exhibits that for WBC parameters, there was no statistically significant difference observed in total leukocyte count (cells/mm^3), ($F_{(20,52)} = 0.94$, p = .539), neutrophil (%), ($F_{(20,52)} = 1.40$, p = .162), absolute neutrophil count, ($F_{(20,52)} = 0.98$, p = .495), lymphocyte (%), ($F_{(20,52)} = 1.51$, p = .117), absolute lymphocyte count, ($F_{(20,52)} = 1.54$, p = .105), monocyte (%), ($F_{(20,52)} = 0.67$, p = .513), eosinophil (%), ($F_{(20,52)} = 0.43$, p = .979), and basophil (%), ($F_{(20,52)} = 0.60$, p = .373), except for platelet count (10^6 cells/mm^3), ($F_{(20,52)} = 2.8$,

**Table 7. Distribution of organism based on birth weight.**

| Micro-organisms | ELBW (n) | LBW (n) | NBW (n) |
|---|---|---|---|
| **Gram-Positive** | | | |
| Candida species | 1 | 1 | 0 |
| Coagulase negative Staphylococcus | 0 | 1 | 5 |
| Enterococcus faecalis | 0 | 1 | 1 |
| Enterococcus faecium | 1 | 0 | 1 |
| Micrococci | 1 | 0 | 3 |
| Staphylococcus aureus | 0 | 0 | 1 |
| Staphylococcus epidermidis | 0 | 1 | 2 |
| Staphylococcus haemolyticus | 1 | 0 | 2 |
| Staphylococcus hominis | 0 | 0 | 1 |
| Staphylococcus saprophyticus | 0 | 0 | 2 |
| Staphylococcus warneri | 0 | 1 | 0 |
| Streptococcus | 0 | 0 | 1 |
| **Gram-Negative** | | | |
| Achromobacter xylosoxidans | 0 | 1 | 0 |
| Acinetobacter baumannii | 3 | 1 | 2 |
| Burkholderia cepacia | 0 | 0 | 1 |
| Elizabethkingia meningoseptica | 3 | 0 | 0 |
| Enterobacter aerogenes | 0 | 0 | 1 |
| Enterobacter cloacae | 2 | 1 | 0 |
| Escherichia coli | 0 | 2 | 0 |
| Klebsiella pneumoniae | 6 | 4 | 9 |
| Pseudomonas aeruginosa | 2 | 1 | 2 |
| Serratia marcescens | 2 | 2 | 0 |
| **TOTAL** | 22 | 17 | 34 |

ELBW — Extremely Low Birth Weight (<1.5 kg)

LBW — Low Birth Weight (1.5-2.5 Kg)

NBW — Normal Birth Weight (>2.5 kg)

**Table 8. ANOVA results of RBC parameters among neonatal septicemia cases.**

| Hematological profile (RBC) | F value | P value | Significance |
|---|---|---|---|
| Hb(g/dl) | 1.069 | 0.408 | Not significant |
| RBC($*10^{12}$/L) | 1.128 | 0.352 | Not significant |
| Hct(%) | 1.11 | 0.369 | Not significant |
| MCV(fl) | 0.601 | 0.894 | Not significant |
| MCH(pg) | 0.55 | 0.929 | Not significant |
| MCHC(g/dl) | 1.269 | 0.242 | Not significant |
| Red Cell Distribution | 1.651 | 0.075 | Not significant |

*F-value*: The ratio of variance between groups to variance within groups.

*P-value*: Value < 0.05 indicates statistical significance.

p = .001), which was statistically very highly significant, suggesting notable variation in platelet levels across different age groups of sepsis onset.

Table 10 shows logistic regression analysis of predictors for term and preterm delivery which suggests that birth weight is the most critical predictor of preterm/term delivery with

**Table 9. ANOVA results of WBC parameters among neonatal septicemia cases.**

| Hematological profile (WBC) | F | P value | Significance |
|---|---|---|---|
| TLC (cells/mm³) | 0.944 | 0.539 | Not significant |
| Neutrophil (%) | 1.406 | 0.162 | Not significant |
| ANC | 0.984 | 0.495 | Not significant |
| Lymphocyte (%) | 1.513 | 0.117 | Not significant |
| ALC | 1.546 | 0.105 | Not significant |
| Monocyte (%) | 0.967 | 0.513 | Not significant |
| Eosinophil (%) | 0.431 | 0.979 | Not significant |
| Basophil (%) | 1.106 | 0.373 | Not significant |
| Platelet Count (10⁶cells/mm³) | 2.825 | 0.001 | Highly significant |

*F-value*: *The ratio of variance between groups to variance within groups.*

*P-value*: *Value < 0.05 indicates statistical significance.*

**Table 10. Logistic Regression Analysis of Predictors for Term and Preterm Delivery.**

| Variable | B (Coefficient) | p-value | Odds Ratio | 95% CI | Significance |
|---|---|---|---|---|---|
| **Birth weight** | 3.488 | < 0.001 | 32.730 | (8.15 -131.32) | Significant |
| **TLC (cells/mm³)** | 0.000 | 0.168 | 1.000 | (1.00 - 1.00) | Not significant |
| **Neutrophil (%)** | 0.039 | 0.103 | 1.039 | (0.993 - 1.087) | Not significant |

B **(Coefficient):** *Indicates how much the dependent variable is expected to change for a one-unit increase in the predictor variable.*

*P-value*: *Value < 0.05 indicates statistical significance*

a very strong and statistically significant effect ($p < 0.05$) but TLC and neutrophil percentage (as an important diagnostic marker for bacterial infection) show a weak and non-significant relationship with preterm/term delivery.

## Discussion

Neonatal sepsis remains a significant cause of morbidity and mortality globally, particularly affecting infants in NICU. This research is a retrospective cross-sectional analysis that aims to examine hematological indicators in newborns with sepsis. The aim is to provide diagnostic insights related to this condition. Neonatal sepsis does not present with a specific clinical presentation, and it remains challenging to diagnose it to prevent unwanted antibiotic usage. Currently, the early diagnosis of neonatal septicemia is based primarily on clinical evaluation. However, many neonates are treated with several days of antibiotics because of a possible infection while waiting for a blood culture report. This results in a high number of neonates without evident septicemia being treated with antibiotics.

Studies on indirect infection markers demonstrate that the hematological blood profile is reliable and helpful in diagnosing neonatal septicemia earlier. The competency of the hematological lab parameters in neonatal sepsis has been found to vary in the literature. In a study on the role of various hematological parameters in the diagnosis of clinically suspected cases of neonatal septicemia done by Gautam et al. [22], 100 neonates were evaluated for sepsis based on the clinical history and signs and symptoms that were present at the time of admission at the neonatal intensive care unit.

Neonatal sepsis is classified into two categories based on the timing of symptom onset: early-onset sepsis (EOS), occurring within the first 72 hours of life, and late-onset sepsis (LOS), manifesting after 72 hours. In this study, the highest incidence of sepsis was observed on day 0, accounting for 23.3% of cases, followed by day 5 with 11% of cases. This

predominance of early-onset cases suggests a significant contribution of congenital or perinatal factors to neonatal sepsis. The findings of present study align with previous research indicating that EOS is often associated with pathogens acquired during the birthing process, such as Group B Streptococcus and Escherichia coli [23]. However, the relatively high incidence observed on day 5 warrants further investigation, as it may indicate a transition period where both perinatal and environmental factors contribute to sepsis risk. Contrary to the results of this study, some studies have reported a higher prevalence of LOS, particularly in preterm infants, where environmental exposures and invasive procedures increase susceptibility to infections [24]. This discrepancy may be attributed to differences in study populations, healthcare settings, and infection control practices. The early peak in sepsis cases observed in this study underscores the importance of vigilant monitoring and prompt intervention during the initial days of life. Implementing strategies such as maternal screening for infections, intrapartum antibiotic prophylaxis, and strict aseptic techniques in neonatal care can potentially reduce the incidence of EOS [25].

The relationship between neonatal sepsis and birth weight is particularly evident in infants with very low birth weight (VLBW), who are at increased risk for sepsis due to their underdeveloped immune systems and prolonged hospital stays [26]. In this study, normal birth weight (NBW) infants represented the majority of sepsis cases at 46.6%, followed by very low birth weight (VLBW) infants at 30.1% and low birth weight (LBW) infants at 23.3%. This distribution is somewhat unexpected; as previous research frequently associates a higher susceptibility to sepsis with VLBW infants. Halder et al. (2020) found that VLBW infants experienced a high incidence of sepsis (62%) in their study cohort, highlighting the vulnerability of these infants due to their underdeveloped immune systems and the need for prolonged hospital care [27].

Similarly, Bai et al. (2021) noted that gram-negative organisms predominantly caused infections in LBW preterm infants, who often face more severe infections than NBW infants [26]. This distribution aligns with findings from Kanwal et al. (2024), who found no significant relationship between birth weight and neonatal sepsis in their Karachi-based study, where half of the sepsis cases were in neonates of healthy weights (2.1–3 kg) [28]. This suggests that, although VLBW infants are often at a higher risk due to their immature immune systems, sepsis can also impact infants with higher birth weights.

The TLC was positive in 30 cases; 41 cases showed positive ANC, and 28 cases showed a positive platelet value. In the present study, the total leukocyte count remained positive in 23.3% of cases, while the ANC count remained normal in most cases. Moreover, in the study done by Gautam et al. [22], out of 76 blood culture-positive cases, 25 cases showed growth of Klebsiella sp., 12 cases showed growth of Acinetobacter sp., and 8 cases showed growth of Coagulase-negative staphylococcus. In present study, Klebsiella was the commonest pathogen, documented in about 19 cases out of 73 culture-proven sepsis, most of them present in early-onset sepsis, especially in neonates with extremely low birth weight, followed by Acinetobacter baumannii and CONS in about 6 cases each. Furthermore, 41 of the cases were primarily late-onset sepsis.

The investigations of this study discovered many significant hematological changes linked to neonatal sepsis. Leukocyte abnormalities were common, with considerable departures from normal ranges seen in neutrophil and lymphocyte counts. A significant number of patients had neutropenia, which is a recognized indicator of the severity of sepsis. This is consistent with earlier studies that emphasize neutropenia as an important indicator for identifying newborns who are at risk of severe septicemia [29]. The TLC gave an image of the underlying infection of 83.5% in neonatal sepsis, which correlates with Elsayed et al.'s literature [30].

The absolute neutrophil count played an important role in assessing the infection status in infants, but it remained almost normal in this study. Neutrophils and lymphocyte

abnormalities of 85.3% and 86.3% were noted, respectively. Neonatal sepsis showed a male predominance of 63% in present study, which is in accordance with Jimba Jatsho et al.'s 57.3% [31]. The other WBC parameters, such as basophil, monocyte, and eosinophil, were near the normal range in most septicemia cases. Furthermore, thrombocytopenia was identified in roughly a quarter of patients, underscoring its importance as a marker of coagulation dysfunction and the probable development of disseminated intravascular coagulation (DIC) [32].

The decreased platelet count was seen in 23.3% of cases, identifying the sepsis cases, which correlates to the study on thrombocytopenia in neonatal sepsis by Suzanne et al. [33], which showed thrombocytopenia occurred in 20% of septic neonates. Although not extensively addressed in the research, anemia in this group, including inflammation-induced inhibition of erythropoiesis [34], might worsen the clinical course of septic newborns owing to its multifactorial etiology. The hematological measures studied, including RBC indices and hemoglobin levels, gave insights into the greater systemic effect of sepsis on newborn physiology. The hematocrit was deranged in 50% of cases, especially in preterm infants, which conflicts with Rodwell et al.'s [30] literature.

This research highlighted a diverse microbial landscape linked to newborn sepsis, largely containing gram-negative organisms such as Klebsiella pneumoniae, followed by gram-positive bacteria like Staphylococcus aureus and fungal species like Candida. This distribution parallels results from previous worldwide research, demonstrating the diverse epidemiology of newborn sepsis and the significance of customized antibiotic regimens based on local susceptibility patterns [35].

The diagnostic issues offered by newborn sepsis were addressed in light of the existing limits of blood culture, the gold standard for diagnosis, but it is time-consuming and can lead to inaccurate results due to various factors, such as antibiotic treatment prior to sampling or contamination, and it should not be used solely for the confirmation of the sepsis case in neonates. Despite its specificity, blood culture suffers from low sensitivity and longer turnaround times, prompting the development of complementary diagnostic techniques such as biomarkers (e.g., C-reactive protein, Procalcitonin) and hematological indices (e.g., neutrophil-to-lymphocyte ratio) [36].

These indicators, although useful, need additional confirmation in newborn populations to maximize their value in early diagnosis and therapy.

The results underline the important need for rapid and reliable identification of newborn sepsis to reduce unfavorable outcomes such as septic shock and multiorgan failure. Advances in rapid diagnostic technologies, particularly molecular assays and point-of-care testing, show promise for enhancing diagnostic accuracy and accelerating treatment interventions in newborns. Such technologies have demonstrated their potential in addressing emerging public health threats. This is supported by other study results on the application of molecular assays and next-generation sequencing in diagnosing and managing emerging infectious diseases, highlighting the importance of novel diagnostic tools for infectious diseases. These advancements emphasize the need for integrating rapid diagnostic techniques into routine clinical practice to improve neonatal care outcomes [37,38]. Additionally, specialized research concentrating on host-response biomarkers and microbial genetics may further improve diagnostic algorithms and increase individualized treatment options. This perspective aligns with findings highlighting the importance of host immune modulation and pathogen-specific molecular pathways in the development of novel diagnostic strategies [39].

## Conclusion

In the diagnostic assessment of newborn sepsis, blood culture has a higher positivity rate and is the gold standard diagnostic method. However, several factors may affect the results, such as contamination, antibiotic treatment prior to culture testing, and a delay in reporting. In

this study, hematocrit derangement, TLC, MCV, and platelet count changes give early signs and, therefore, can be used as a diagnostic tool for early diagnosis of neonatal septicemia but altogether give a better diagnosis of septicemia, which helps for the appropriate management and antibiotic therapy.

## Acknowledgment

The authors would like to express their thanks and gratitude to AlMaarefa University, Riyadh for providing the support for this study.

## Author contributions

**Conceptualization:** Jeivanth S.B., Shreemathee Baskar, K. Santhosh Kumar.

**Data curation:** Jeivanth S.B., Shreemathee Baskar, Mohammad Fareed, Osama Elshahat Mostafa, Amen Bawazir, Khalid I. AlQumaizi.

**Formal analysis:** Jeivanth S.B., Shreemathee Baskar, Mohammad Fareed, K. Santhosh Kumar, Osama Elshahat Mostafa.

**Funding acquisition:** Mohammad Fareed, Amen Bawazir, Khalid I. AlQumaizi.

**Investigation:** Jeivanth S.B., Shreemathee Baskar.

**Methodology:** Jeivanth S.B., Shreemathee Baskar, Mohammad Fareed, Khalid I. AlQumaizi.

**Project administration:** Shreemathee Baskar.

**Resources:** Shreemathee Baskar, K. Santhosh Kumar, Amen Bawazir, Khalid I. AlQumaizi.

**Software:** Jeivanth S.B., Mohammad Fareed, K. Santhosh Kumar, Osama Elshahat Mostafa.

**Supervision:** Mohammad Fareed.

**Validation:** Shreemathee Baskar, K. Santhosh Kumar, Osama Elshahat Mostafa, Khalid I. AlQumaizi.

**Visualization:** Jeivanth S.B., K. Santhosh Kumar, Amen Bawazir, Khalid I. AlQumaizi.

**Writing – original draft:** Jeivanth S.B., Shreemathee Baskar.

**Writing – review & editing:** Jeivanth S.B., Shreemathee Baskar, Mohammad Fareed, Osama Elshahat Mostafa, Amen Bawazir, Khalid I. AlQumaizi.

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
