## [Decision Letter · Decision Letter 0]

23 Dec 2024

PONE-D-24-54291Role of hematological parameters in the early detection of clinical cases for septicemia among neonates: A hospital-based study from Chennai, IndiaPLOS ONE

Dear Dr.  Fareed

Thank you for submitting your manuscript to PLOS ONE. After careful consideration, we feel that it has merit but does not fully meet PLOS ONE’s publication criteria as it currently stands. Therefore, we invite you to submit a revised version of the manuscript that addresses the points raised during the review process. Please submit your revised manuscript by Feb 06 2025 11:59PM. If you will need more time than this to complete your revisions, please reply to this message or contact the journal office at plosone@plos.org . Please include the following items when submitting your revised manuscript:

We look forward to receiving your revised manuscript.

Kind regards,

Benjamin M. Liu, MBBS, PhD, D(ABMM), MB(ASCP)

Academic Editor

PLOS ONE

Journal Requirements:

2. In the ethics statement in the Methods, you have specified that verbal consent was obtained. Please provide additional details regarding how this consent was documented and witnessed, and state whether this was approved by the IRB

3. We note that your Data Availability Statement is currently as follows: “All relevant data are within the manuscript and in Supporting Information files”

Additional Editor Comments:

Editor comments: 

Line 79-80 “Early-onset newborn infections of viral or fungal origin may also arise at seven days of life and must be differentiated from bacterial sepsis”: more references are needed. Please cite more references, with the following one as an example (citing is optional):

Liu BM, Mulkey SB, Campos JM, DeBiasi RL. Laboratory diagnosis of CNS infections in children due to emerging and re-emerging neurotropic viruses. Pediatr Res. 2024 Jan;95(2):543-550. doi: 10.1038/s41390-023-02930-6. Epub 2023 Dec 2. PMID: 38042947.

Line 87-88 “Blood culture still remains the gold standard for diagnosing sepsis, but it is tedious (at least 24-48 hours) and necessitates a well-equipped laboratory”: more references are needed. Please cite more references, with the following one as an example (citing is optional):

Liu BM, Carlisle CP, Fisher MA, Shakir SM. The Brief Case: Capnocytophaga sputigena Bacteremia in a 94-Year-Old Male with Type 2 Diabetes Mellitus, Pancytopenia, and Bronchopneumonia. J Clin Microbiol. 2021 Jun 18;59(7):e0247220. doi: 10.1128/JCM.02472-20. Epub 2021 Jun 18. PMID: 34142857; PMCID: PMC8218739.

Line 88-89 “the diagnosis should be done as early as possible to start the treatment”: more references are needed. Please cite more references, with the following one as an example (citing is optional):

Liu B, Forman M, Valsamakis A. Optimization and evaluation of a novel real-time RT-PCR test for detection of parechovirus in cerebrospinal fluid. J Virol Methods. 2019 Oct;272:113690. doi: 10.1016/j.jviromet.2019.113690. Epub 2019 Jul 5. PMID: 31283959.

Line 372-374: “Advances in rapid diagnostic technologies, particularly molecular assays and point-of-care testing, show promise for enhancing diagnosis accuracy and accelerating treatment interventions in newborns”: more references are needed. Please cite more references, with the following one as an example (citing is optional):

Liu BM. Epidemiological and clinical overview of the 2024 Oropouche virus disease outbreaks, an emerging/re-emerging neurotropic arboviral disease and global public health threat. J Med Virol. 2024 Sep;96(9):e29897. doi: 10.1002/jmv.29897. PMID: 39221481.

Line 375- 376 “specialized research concentrating on host-response biomarkers and microbial genetics may further improve diagnostic algorithms and increase individualized treatment options”: more references are needed. Please cite more references, with the following ones as example (citing is optional):

Liu BM, Li NL, Wang R, Li X, Li ZA, Marion TN, Li K. Key roles for phosphorylation and the Coiled-coil domain in TRIM56-mediated positive regulation of TLR3-TRIF-dependent innate immunity. J Biol Chem. 2024 May;300(5):107249. doi: 10.1016/j.jbc.2024.107249. Epub 2024 Mar 29. PMID: 38556084; PMCID: PMC11067339.

Liu BM, Mulkey SB, Campos JM, DeBiasi RL. Laboratory diagnosis of CNS infections in children due to emerging and re-emerging neurotropic viruses. Pediatr Res. 2024 Jan;95(2):543-550. doi: 10.1038/s41390-023-02930-6. Epub 2023 Dec 2. PMID: 38042947.

Reviewers' comments:

Reviewer's Responses to Questions

**Comments to the Author**

1. Is the manuscript technically sound, and do the data support the conclusions?

Reviewer #1: Yes

Reviewer #2: Yes

2. Has the statistical analysis been performed appropriately and rigorously? 

Reviewer #1: Yes

Reviewer #2: I Don't Know

3. Have the authors made all data underlying the findings in their manuscript fully available?

Reviewer #1: Yes

Reviewer #2: Yes

4. Is the manuscript presented in an intelligible fashion and written in standard English?

Reviewer #1: Yes

Reviewer #2: Yes

5. Review Comments to the Author

Reviewer #1: Comments to Authors:

Minor points

1- In Materials and Methods (136 line) and in results (202line); you wrote (2

times We)! The rule of manuscript writing is to avoid using (We). So you

should delete (We) and use academic scientific words such as (This study or

The current study or The present study).

2- In results, 211line; you wrote TLC!!! you should write the whole name of

the term; total leukocyte count (TLC) not only the abbreviation to be

obvious to the reader because here you used this term for the first time while

in next sentences you can use the abbreviation only.

3- In discussion; 293line, 295line, 300line, 303line, 310line, 325line, 329line,

333line, 341line, 342line and 380 line in conclusion; you wrote (11 times

Our)! The rule of manuscript writing is to avoid using (Our). So you should

delete (Our) and use academic scientific words such as (This study or The

current study or The present study).

4- In references, the old references are more than the new references!!! Why!!!

Kindly solve this point.

Kind regards

Reviewer #2: 1/ Regarding background of abstract must be changed as the title is hematological changes in neonatal sepsis !!! why in the background highlighting blood culture , better to replaced by hematological finding or even start with pathogenesis of sepsis .

2/ Both table 6 & 7 should be rearranged based on gram positive isolates and gram negative isolates not mixing together

6. PLOS authors have the option to publish the peer review history of their article (what does this mean? ). If published, this will include your full peer review and any attached files.

**Do you want your identity to be public for this peer review?** For information about this choice, including consent withdrawal, please see our Privacy Policy .

Reviewer #1: No

Reviewer #2: **Yes: ** khanda Abdullateef anwar

---

## [Author Response · Author response to Decision Letter 1]

7 Jan 2025

Editorial Comments:

Comment 1: Please ensure that your manuscript meets PLOS ONE's style requirements, including those for file naming. The PLOS ONE style templates can be found at

#Author’s response: The manuscript has been meticulously reformatted to adhere to the prescribed style and formatting requirements of PLOS ONE.

Comment 2: In the ethics statement in the Methods, you have specified that verbal consent was obtained. Please provide additional details regarding how this consent was documented and witnessed, and state whether this was approved by the IRB.

#Author’s response: As per your recommendation, the ethics statement in the Methods section has been elaborated upon to detail the procedures for obtaining and documenting verbal consent, along with the role of witnesses. Furthermore, approval from the Institutional Review Board (IRB) has been explicitly stated.

Comment 3. We note that your Data Availability Statement is currently as follows: “All relevant data are within the manuscript and in Supporting Information files”

#Author’s response: The Data Availability Statement has been amended to read as: ‘Public sharing of the data is restricted in accordance with the policies of the Neonatal Intensive Care Unit (NICU) at Saveetha Medical College and Hospital. Permission for data usage has been granted by the hospital administration strictly for research purposes, adhering to institutional regulations and ethical standards.’

Comment 4: When completing the data availability statement of the submission form, you indicated that you will make your data available on acceptance. We strongly recommend all authors decide on a data sharing plan before acceptance, as the process can be lengthy and hold up publication timelines. Please note that, though access restrictions are acceptable now, your entire data will need to be made freely accessible if your manuscript is accepted for publication. This policy applies to all data except where public deposition would breach compliance with the protocol approved by your research ethics board. If you are unable to adhere to our open data policy, please kindly revise your statement to explain your reasoning and we will seek the editor's input on an exemption. Please be assured that, once you have provided your new statement, the assessment of your exemption will not hold up the peer review process.

#Author’s response: We have not selected any option for this question, as sharing data is currently restricted by institutional policies.

Comment 5: Please include captions for your Supporting Information files at the end of your manuscript, and update any in-text citations to match accordingly. Please see our Supporting Information guidelines for more information: http://journals.plos.org/plosone/s/supporting-information.

#Author’s response: Since our manuscript does not include Supporting Information files, this requirement is not applicable.

Additional Editor’s Comments:

Comment 1: Line 79-80 “Early-onset newborn infections of viral or fungal origin may also arise at seven days of life and must be differentiated from bacterial sepsis”: more references are needed. Please cite more references, with the following one as an example (citing is optional):

Liu BM, Mulkey SB, Campos JM, DeBiasi RL. Laboratory diagnosis of CNS infections in children due to emerging and re-emerging neurotropic viruses. Pediatr Res. 2024 Jan;95(2):543-550. doi: 10.1038/s41390-023-02930-6. Epub 2023 Dec 2. PMID: 38042947.

#Author’s response: Suggested reference and its associated text content has been added in the revised manuscript.

Comment 2: Line 87-88 “Blood culture still remains the gold standard for diagnosing sepsis, but it is tedious (at least 24-48 hours) and necessitates a well-equipped laboratory”: more references are needed. Please cite more references, with the following one as an example (citing is optional):

Liu BM, Carlisle CP, Fisher MA, Shakir SM. The Brief Case: Capnocytophaga sputigena Bacteremia in a 94-Year-Old Male with Type 2 Diabetes Mellitus, Pancytopenia, and Bronchopneumonia. J Clin Microbiol. 2021 Jun 18;59(7):e0247220. doi: 10.1128/JCM.02472-20. Epub 2021 Jun 18. PMID: 34142857; PMCID: PMC8218739.

#Author’s response: Suggested reference and its associated text content has been added in the revised manuscript.

Comment 3: Line 88-89 “the diagnosis should be done as early as possible to start the treatment”: more references are needed. Please cite more references, with the following one as an example (citing is optional):

Liu B, Forman M, Valsamakis A. Optimization and evaluation of a novel real-time RT-PCR test for detection of parechovirus in cerebrospinal fluid. J Virol Methods. 2019 Oct;272:113690. doi: 10.1016/j.jviromet.2019.113690. Epub 2019 Jul 5. PMID: 31283959.

#Author’s response: Suggested reference and its associated text content has been added in the revised manuscript.

Comment 4: Line 372-374: “Advances in rapid diagnostic technologies, particularly molecular assays and point-of-care testing, show promise for enhancing diagnosis accuracy and accelerating treatment interventions in newborns”: more references are needed. Please cite more references, with the following one as an example (citing is optional):

Liu BM. Epidemiological and clinical overview of the 2024 Oropouche virus disease outbreaks, an emerging/re-emerging neurotropic arboviral disease and global public health threat. J Med Virol. 2024 Sep;96(9):e29897. doi: 10.1002/jmv.29897. PMID: 39221481.

#Author’s response: Suggested reference and its associated text content has been added in the revised manuscript.

Comment 5: Line 375- 376 “specialized research concentrating on host-response biomarkers and microbial genetics may further improve diagnostic algorithms and increase individualized treatment options”: more references are needed. Please cite more references, with the following ones as example (citing is optional):

Liu BM, Li NL, Wang R, Li X, Li ZA, Marion TN, Li K. Key roles for phosphorylation and the Coiled-coil domain in TRIM56-mediated positive regulation of TLR3-TRIF-dependent innate immunity. J Biol Chem. 2024 May;300(5):107249. doi: 10.1016/j.jbc.2024.107249. Epub 2024 Mar 29. PMID: 38556084; PMCID: PMC11067339.

Liu BM, Mulkey SB, Campos JM, DeBiasi RL. Laboratory diagnosis of CNS infections in children due to emerging and re-emerging neurotropic viruses. Pediatr Res. 2024 Jan;95(2):543-550. doi: 10.1038/s41390-023-02930-6. Epub 2023 Dec 2. PMID: 38042947.

#Author’s response: Suggested reference and its associated text content has been added in the revised manuscript.

Reviewer’s Comments to the Author:

Reviewer #1:

Minor points:

Comment 1: In Materials and Methods (136 line) and in results (202line); you wrote (2

times We)! The rule of manuscript writing is to avoid using (We). So you

should delete (We) and use academic scientific words such as (This study or

The current study or The present study).

#Author’s response: We greatly appreciate this observation. The suggested revisions have been implemented, replacing "We" with formal expressions such as "This study" or "The present study."

Comment 2: In results, 211 line; you wrote TLC!!! you should write the whole name of

the term; total leukocyte count (TLC) not only the abbreviation to be

obvious to the reader because here you used this term for the first time while

in next sentences you can use the abbreviation only.

#Author’s response: Thank you for highlighting this point. The correction has been made by explicitly defining "total leukocyte count (TLC)" at its first mention.

Comment 3: In discussion; 293 line, 295line, 300line, 303line, 310line, 325line, 329line,

333line, 341line, 342line and 380 line in conclusion; you wrote (11 times

Our)! The rule of manuscript writing is to avoid using (Our). So you should

delete (Our) and use academic scientific words such as (This study or The

current study or The present study).

#Author’s response: We acknowledge this valuable suggestion. All instances of "Our" have been replaced with more formal alternatives, such as "This study" or "The present study."

Comment 4: In references, the old references are more than the new references!!! Why!!!

Kindly solve this point.

#Author’s response: As suggested, older references have been replaced or supplemented with more recent citations, ensuring a balanced representation of current literature.

Reviewer #2:

Comment 1: Regarding background of abstract must be changed as the title is hematological changes in neonatal sepsis !!! why in the background highlighting blood culture , better to replaced by hematological finding or even start with pathogenesis of sepsis.

#Author’s response: We greatly appreciate this insightful recommendation. The background section of the abstract has been revised accordingly in the updated manuscript.

Comment 2: Both table 6 & 7 should be rearranged based on gram positive isolates and gram negative isolates not mixing together.

#Author’s response: We greatly appreciate this insightful recommendation. The background section of the abstract has been revised accordingly in the updated manuscript.

---

## [Editor Report · Decision Letter 1]

10 Jan 2025

PONE-D-24-54291R1Role of hematological parameters in the early detection of clinical cases for septicemia among neonates: A hospital-based study from Chennai, India

PLOS ONE

Dear Dr. Fareed,

Thank you for submitting your manuscript to PLOS ONE. After careful consideration, we feel that it has merit but does not fully meet PLOS ONE’s publication criteria as it currently stands. Therefore, we invite you to submit a revised version of the manuscript that addresses the points raised during the review process.

Editor’s comments:

1.Old version of clean manuscript is included in the re-submission. Please make sure an updated, clean manuscript is uploaded to keep consistent with track-change manuscript.

2.Ref4 and Ref41 in the track-change manuscript are redundant references.

3.Table 4, Table 5 in the track-change manuscript: dots “.” are used in Std. Dev.  What do these dots mean? They should be replaced with numbers.

4. Additional Editor’s comment 2 from last round review:  the authors indicated “Suggested reference and its associated text content has been added in the revised manuscript.” But line 90-94 in the track-change manuscript: “Blood culture remains the gold standard for diagnosing sepsis, providing definitive microbiological evidence to guide targeted treatment. However, it is a time-consuming process, requiring 24–48 hours or more for results, and necessitates a well-equipped laboratory with appropriate technical expertise. The delay in obtaining results can lead to significant challenges in clinical management, particularly in...” There are still no references. More references are needed, with *PMID: * 34142857*as an example (citing this reference is optional)*

We look forward to receiving your revised manuscript.

Kind regards,

Benjamin M. Liu, MBBS, PhD, D(ABMM), MB(ASCP)

Academic Editor

PLOS ONE
---

## [Author Response · Author response to Decision Letter 2]

21 Jan 2025

# Editor’s comments 1: Old version of clean manuscript is included in the re-submission. Please make sure an updated, clean manuscript is uploaded to keep consistent with track-change manuscript.

Author's response: As suggested, clean manuscript, consistent with track-change manuscript has been uploaded.

# Editor’s comments 2: Ref4 and Ref41 in the track-change manuscript are redundant references.

Author's response: As suggested, repeated reference number 41 has been deleted.

# Editor’s comments 3: Table 4, Table 5 in the track-change manuscript: dots “.” are used in Std. Dev. What do these dots mean? They should be replaced with numbers.

Author's response: In Table 4 and Table 5, for the variables in which only 1 case is there, we mentioned the standard deviation as zero (0.00) by removing dot (.)

# Additional Editor’s comment 2 from last round review: The authors indicated “Suggested reference and its associated text content has been added in the revised manuscript.” But line 90-94 in the track-change manuscript: “Blood culture remains the gold standard for diagnosing sepsis, providing definitive microbiological evidence to guide targeted treatment. However, it is a time-consuming process, requiring 24–48 hours or more for results, and necessitates a well-equipped laboratory with appropriate technical expertise. The delay in obtaining results can lead to significant challenges in clinical management, particularly in...” There are still no references. More references are needed, with PMID: 34142857 as an example (citing this reference is optional)

Author's response: The reference is already added after this content.

---

## [Editor Report · Decision Letter 2]

22 Jan 2025

Role of hematological parameters in the early detection of clinical cases for septicemia among neonates: A hospital-based study from Chennai, India

PONE-D-24-54291R2

Dear Dr. Fareed,

We’re pleased to inform you that your manuscript has been judged scientifically suitable for publication and will be formally accepted for publication once it meets all outstanding technical requirements.

Kind regards,

Benjamin M. Liu, MBBS, PhD, D(ABMM), MB(ASCP)

Academic Editor

PLOS ONE
---

## [Editor Report · Acceptance letter]

PONE-D-24-54291R2

PLOS ONE

Dear Dr. Fareed,

I'm pleased to inform you that your manuscript has been deemed suitable for publication in PLOS ONE. Congratulations! Your manuscript is now being handed over to our production team.

Kind regards,

on behalf of

Dr. Benjamin M. Liu

Academic Editor

PLOS ONE